# Downregulation of Dystrophin Expression Occurs across Diverse Tumors, Correlates with the Age of Onset, Staging and Reduced Survival of Patients

**DOI:** 10.3390/cancers15051378

**Published:** 2023-02-21

**Authors:** Nancy Alnassar, Malgorzata Borczyk, Georgia Tsagkogeorga, Michal Korostynski, Namshik Han, Dariusz C. Górecki

**Affiliations:** 1School of Pharmacy and Biomedical Sciences, University of Portsmouth, Portsmouth PO1 2DT, UK; 2Laboratory of Pharmacogenomics, Maj Institute of Pharmacology PAS, 31155 Krakow, Poland; 3Milner Therapeutics Institute, University of Cambridge, Cambridge CB4 0WS, UK; 4STORM Therapeutics Ltd., Babraham Research Campus, Cambridge CB22 3AT, UK; 5Cambridge Centre for AI in Medicine, University of Cambridge, Cambridge CB2 0QQ, UK

**Keywords:** DMD, duchenne muscular dystrophy, dystrophin, malignancy, cancer, sarcoma

## Abstract

**Simple Summary:**

Mutations of the *DMD* gene, encoding dystrophins, cause Duchenne muscular dystrophy (DMD). We found that while *DMD* transcription occurs throughout a spectrum of normal tissues, it is frequently downregulated across various malignancies. The molecular signature associated with this downregulation matches transcriptomic changes found in Duchenne muscles, even though most of these malignancies originate from tissues never previously associated with dystrophin expression or function. Importantly, we found that reduced *DMD* expression across different tumors was associated with reduced patients’ survival and higher tumor stage. These findings call for re-evaluation of the current view that dystrophin expression found across numerous tissues is the result of an “illegitimate transcription” and demonstrate that the significance of this gene goes beyond its known involvement in Duchenne muscular dystrophy. Moreover, these data unite and explain the growing evidence that the *DMD* gene has a role in tumors.

**Abstract:**

Altered dystrophin expression was found in some tumors and recent studies identified a developmental onset of Duchenne muscular dystrophy (DMD). Given that embryogenesis and carcinogenesis share many mechanisms, we analyzed a broad spectrum of tumors to establish whether dystrophin alteration evokes related outcomes. Transcriptomic, proteomic, and mutation datasets from fifty tumor tissues and matching controls (10,894 samples) and 140 corresponding tumor cell lines were analyzed. Interestingly, dystrophin transcripts and protein expression were found widespread across healthy tissues and at housekeeping gene levels. In 80% of tumors, *DMD* expression was reduced due to transcriptional downregulation and not somatic mutations. The full-length transcript encoding Dp427 was decreased in 68% of tumors, while Dp71 variants showed variability of expression. Notably, low expression of dystrophins was associated with a more advanced stage, older age of onset, and reduced survival across different tumors. Hierarchical clustering analysis of *DMD* transcripts distinguished malignant from control tissues. Transcriptomes of primary tumors and tumor cell lines with low *DMD* expression showed enrichment of specific pathways in the differentially expressed genes. Pathways consistently identified: ECM-receptor interaction, calcium signaling, and PI3K-Akt are also altered in DMD muscle. Therefore, the importance of this largest known gene extends beyond its roles identified in DMD, and certainly into oncology.

## 1. Introduction

Duchenne muscular dystrophy (DMD) is a debilitating and lethal neuromuscular disorder. Diagnosis is made between the age of two and five, but transcriptomes from pre-symptomatic DMD patients reveal typical dystrophic abnormalities [1]. In fact, studies of human fetuses [2,3,4] and in various animal models [5,6,7] revealed that the pathology starts during prenatal development and continues into adulthood. The first DMD defects are detectable in developing cells even before their differentiation into muscle [8]. Given that muscle regeneration replicates processes occurring in muscle development, and that some developmental mechanisms are reactivated in tumors, it is intriguing that changes in *DMD* gene expression are increasingly being described in various malignancies [9]. Moreover, *DMD* downregulation affects cell proliferation [10,11,12], adhesion, migration, and invasion [12]—traits that are commonly associated with tumor development. Gaining some understanding of whether these alterations, in a range of diverse cells, have a common origin and lead to a related outcome would expand our knowledge of the role of the *DMD* gene and potentially open up new treatment avenues.

Such analysis must take into consideration the complexity of DMD, the largest human gene known, with 79 exons and eight independent tissue-specific promoters driving the expression of distinct dystrophin isoforms. Three promoters control the expression of 14-kb full-length transcripts encoding 427 kDa isoforms (Dp427). The Dp427m is expressed in myofibers and muscle stem (satellite) cells. In myofibers, it interacts with the dystrophin-associated protein complex (DAPC) [13] with structural and scaffolding roles and an involvement in the regulation of various signaling pathways [14]. Yet, in satellite cells, Dp427m has a different interactome and it is essential for asymmetric cell divisions [11,15]; in myoblasts the loss of its expression results in abnormalities of cell proliferation and migration [10,16,17]. Dp427c and Dp427p full-length dystrophins are expressed in various neurons [18,19], where their loss during development has been linked to the neuropsychological impairment in DMD.

Moreover, intragenic promoters give rise to transcripts encoding truncated isoforms: Dp260 in the retina, Dp140 in the CNS, Dp116 in Schwann glia [20], and Dp71, which is the most ubiquitous *DMD* product. Alternative splicing adds further structural and functional diversity [21,22]. *DMD* transcripts are summarized in Appendix A.

Importantly, loss of Dp427 expression, which is both necessary and sufficient for the occurrence of Duchenne MD, was also associated with increased metastasis of tumors with myogenic programs [17] and soft tissue sarcomas [23]. In contrast to this potential tumor suppressor role of the full-length dystrophin, Dp71 expression was essential for myogenic tumor cell growth [17,23]. Interestingly, a putative role for the *DMD* gene was also suggested in various non-myogenic cancers, including carcinomas [24,25,26], melanoma [27], leukemia [28], lymphoma [24], and CNS tumors [29,30,31].

Yet, many of the tumors showing phenotypic changes associated with decreased *DMD* gene expression originate from healthy tissues that are not generally believed to express the full-length dystrophin protein. Seemingly paradoxical, this observation is not limited to tumors: myoblasts, lymphocytes, and endotheliocytes express 14-kb dystrophin transcripts but are not known to synthesize detectable levels of dystrophin protein. Nevertheless, *DMD* downregulation leads to significant functional abnormalities in these cells [16,32,33].

We hypothesized that malignancy can be used as a model to investigate changes in *DMD* gene expression across normal tissues and corresponding tumors, and aid our understanding of the overall role of this gene, which clearly extends beyond Duchenne MD. Therefore, we studied *DMD* mRNA and protein expression across various normal tissues and matching tumors, and explored transcriptomic alterations in primary tumors and corresponding tumor cell lines with altered *DMD* expression to identify putative downstream molecular pathways that could be associated with *DMD* dysregulation across human tissues. We also examined the association of *DMD* gene expression with the onset and survival endpoints in tumor patients.

## 2. Methods

### 2.1. Gene, Protein, and Clinical Datasets

The RNA-seq data for tumor (n = 7894) and control (n = 2714) tissue samples was obtained from the UCSC Xena Functional Genomics Browser (http://xena.ucsc.edu, accessed on 10 December 2021). The “RSEM norm_count” and “RSEM expected_count” datasets of the TCGA TARGET GTEx cohort were used to examine expression data at the gene and transcript level, respectively, for a variety of cancers in The Cancer Genome Atlas (TCGA) database (https://www.cancer.gov/tcga, accessed on 10 December 2021), different non-diseased tissues in the Genotype-Tissue Expression (GTEx) project (https://gtexportal.org/, accessed on 10 December 2021), and two pediatric hematological malignancies in the TARGET (Therapeutically Applicable Research To Generate Effective Treatments) database (https://ocg.cancer.gov/programs/target, accessed on 20 July 2022). Samples from the TCGA, TARGET, and GTEx have been re-analyzed and processed by the same bioinformatic pipeline to eliminate batch effects resulting from different computational processing [34]. The expression level of *DMD* gene/transcripts was compared between tumor tissues and histologically normal tissues adjacent to the tumors (NATs) from the TCGA (16 comparisons). When expression data for NATs was not available in the TCGA, a comparison was made between *DMD* gene/transcripts expression in tumor samples from the TCGA and healthy tissue samples in the GTEx database (nine comparisons). Database sources in each comparison are stated in Appendix A. The level of *DMD* transcripts was also compared between two TARGET pediatric malignancies and GTEx healthy samples.

The Proteomics DB data resource (https://www.proteomicsdb.org/vue/, accessed on 15 February 2022) was used to examine the expression of the full-length dystrophin protein in normal tissues. This database contains quantitative proteomics data assembled from liquid chromatography tandem-mass-spectrometry (LC-MS/MS) experiments in a variety of normal tissues [35].

Tissue specificity (TS) scores (i.e., the enrichment) of dystrophin protein across a variety of normal tissues were visualized at (https://tsomics.shinyapps.io/RNA_vs_protein/, accessed on 12 October 2022), as well as the correlation between *DMD* gene and protein expression in 32 different tissue types covering all major organs, including skeletal muscle [36].

The database “PaxDb” (Protein Abundances Across Organisms) was used to examine dystrophin protein abundance data across different normal tissues (https://pax-db.org, accessed on 12 October 2022). This database covers a large number of protein abundance datasets in a variety of tissues. For a specific protein of interest, it provides a table of abundances in all available datasets, along with the rank of this protein in the entire detectable proteome [37].

Dystrophin protein expression in normal tissues was also examined using the Human Protein Atlas (https://www.proteinatlas.org/, accessed on 15 February 2022), which contains results from immunohistochemistry assays conducted in a variety of tissues and cell types [38].

Mutation and somatic copy number alteration (SCNA) data for primary tumor samples was derived from the datasets “somatic mutation (SNP and INDEL)—MC3 public version” and “gene-level copy number (gistic2_thresholded)” of the TCGA Pan-cancer (PANCAN) cohort available at the UCSC Xena Browser. The SCNA data was generated by the GISTIC2 algorithm, which generates putative gene copy number specific calls [39]. It should be noted that deep deletions and amplifications are considered as biologically relevant for individual genes by default. Due to purity and ploidy differences between samples, and because these calls are usually not manually reviewed, there may be false positives and false negatives [40,41].

*DMD* gene expression and mutation data for tumor cell lines from the Cancer Cell Line Encyclopedia was obtained from cBioPortal website (https://www.cbioportal.org, accessed on 19 December 2021).

The log_2_ (norm_value + 1) miRNA mature strand expression data was obtained from the dataset “miRNA mature strand expression—Batch effects normalized miRNA data” of the TCGA PANCAN cohort at the UCSC Xena browser.

Data for the American Joint Committee on Cancer (AJCC) pathologic tumor stage and the patient’s age at the initial diagnosis was derived from the dataset “Phenotype—curated clinical data” of the TCGA PANCAN cohort. Tumor samples were allocated into the following age groups: 14–29 years, 30–39, 40–49, 50–59, 60–69, 70–79, and 80 and above years. These groups corresponded with the age groups for GTEx donors obtained from the GTEx portal with two exceptions. The GTEx data did not have any samples from patients older than 79 years old or younger than 20 years old.

The normalized log-transformed RNA-seq data used to compare gene expression in primary tumor samples with low vs. high *DMD* expression in 15 TCGA primary tumors was obtained from the UCSC Xena Browser. Links to database sources can be found in Appendix A.

Expression data for the *DMD* gene in tumor cell lines was obtained from the “Expression 21Q3 Public” dataset using the Cancer Dependency Map (DepMap) portal (https://depmap.org/portal, accessed on 20 July 2022), which includes RNA-seq expression data for 1379 tumor cell lines from 37 lineages [42].

Gene expression microarray data for skeletal muscle samples from DMD patients (n = 12) and healthy donors (n = 11) was obtained from the dataset GSE1004 available at the Gene Expression Omnibus (GEO) from the National Center for Biotechnology Information (https://www.ncbi.nlm.nih.gov/geo/, accessed on 23 May 2022).

Survival data was derived from the dataset “Phenotype—curated clinical data” of the TCGA PANCAN cohort and the dataset “Phenotype—TARGET donor phenotype” of the TARGET PANCAN cohort available at the UCSC Xena Browser.

The STRING database was used to construct dystrophin’s protein-protein interaction (PPI) network [43].

### 2.2. DMD Transcript Identification and Classification

The “RSEM expected_count” dataset of the TCGA TARGET GTEx cohort in the UCSC Xena Browser contains expression data for thirty *DMD* gene transcripts. The Ensembl database was used for transcript annotation and the identification of coding and non-coding transcripts, and protein products in the UniProt database. However, we note that estimates of transcript expression by RSEM may not be 100% accurate. Protein products were then aligned to known dystrophin isoforms in the UniProt database to identify the isoforms with the highest percentage of sequence identity to these protein products. The HMMER web server (https://www.ebi.ac.uk/Tools/hmmer/, accessed on 23 February 2022) was used for the graphical representations of Pfam domains of different dystrophin isoforms.

### 2.3. Statistical Analysis of Differential DMD Gene and Transcript Expression between Tumor Tissues and Corresponding Controls

Statistical analysis of the expression level of the *DMD* gene and transcripts was performed using the UCSC Xena Browser, using the two-tailed Welch’s t-test. The *p*-Values were then adjusted for multiple testing using the Bonferroni correction; each of the obtained *p*-Values was multiplied by the total numbers of tests, and when this adjustment gave a value above 1, the corrected *p*-Values were set at 1. The Bonferroni-corrected *p*-Values were then compared with the overall level of α (α = 0.05). Log-transformed expression data was downloaded from the UCSC Xena Browser and the mean of the *DMD* gene and transcripts expression level for tumor (A) and control (B) samples was calculated. The following formula was then used to calculate Log Fold Change (LogFC): log (A/B) = log (A) − log (B). The UCSC Xena Browser’s instructions can be found at (https://ucsc-xena.gitbook.io/project/faq/advanced-data-and-datasets#how-do-i-calculate-fold-change-fc, accessed on 12 December 2021).

### 2.4. Hierarchical Clustering Analysis of DMD Transcripts

Thirty *DMD* gene transcripts were ranked based on their mean abundance levels in primary tumors and corresponding control tissues, and the top ten highly expressed transcripts were identified. The expression values of the remaining twenty transcripts were collapsed into one value called “other transcripts”. Expression values of the top ten highly expressed *DMD* transcripts as well as the “other transcripts” values were then standardized to a sum of 1 in order to compare the relative expression levels of these transcripts in different tumor and control tissues. An UPGMA hierarchical clustering analysis was then performed using the software R v4.0.4 on an Euclidean distance matrix with the hclust average method. The dendrogram was cut at different levels to yield 3–7 clusters (Appendix A). Six was chosen as the optimal number of clusters as it was the smallest number that provided more than two clusters with around 10 tissues, and higher numbers of clusters yielded additional smaller clusters with less than three tissues. The R script used to perform the analysis is available at (https://github.com/nancyalnassar/DMD-gene-in-cancer/commit/ca99cba54c769c5be760e1f04a09010685647d42, accessed on 15 December 2022). Next, the expression levels of these ten transcripts were compared across three clusters that contained the majority of the analyzed tissues using the Kruskal-Wallis test and Dunn’s multiple comparisons test.

### 2.5. Association between DMD Gene Expression and Mutation Status

Tumor samples from 23 tumor types with available mutation and SCNAs data (n = 6751) were used to perform a univariate general linear model (GLM) analysis to determine whether there is a significant difference in *DMD* gene expression between samples with different mutation/SCNA types and samples with no mutations/SCNAs in the *DMD* gene locus using SPSS statistical software (v.28.0.0.0). Gender was selected as a random factor to account for gender differences between groups. Estimated Marginal (EM) means were calculated, and *p*-Values were adjusted using the Bonferroni correction.

Next, primary tumor samples and tumor cell line samples (n = 921) were ranked according to their level of *DMD* expression, and those at the bottom 30% and top 30% were selected for further analysis. A Chi Square test was used to determine whether there is an association between *DMD* expression level and mutations in coding and non-coding regions, as well as SCNAs in tumor samples and between *DMD* mutations and expression level in tumor cell lines.

### 2.6. Association between DMD Gene Expression and Cancer Stage and Patient’s Age at the Initial Diagnosis

Tumor samples from 18 tumor types with available data for cancer stage, gender, and patient’s age at the initial diagnosis (n = 6118) were classified into four groups based on the cancer stage: I, II, III and IV. A univariate GLM analysis was performed to determine whether there is a significant difference in *DMD* gene expression between these four stages. Gender and age were selected as random factors, and EM means were calculated for the four groups. The *p*-Values were adjusted using the Bonferroni correction.

### 2.7. Identification of Differentially Expressed Genes (DEGs) and miRNAs between Primary Tumor Samples with Low and High Levels of DMD Gene Expression

Gene expression in primary tumor samples with low vs. high levels of *DMD* gene expression from 15 different primary tumors was compared. For each tumor, samples were divided into three groups: Group A (samples at the bottom 33.3% of *DMD* expression), Group B (samples at the top 33.3% of *DMD* expression) and Group C (the remaining samples with medium level of *DMD* expression). Using the software R v4.0.4, an Analysis of Variance (ANOVA) was preformed to identify the DEGs between the three groups, and *p*-Values were adjusted using the False Discovery Rate (FDR) correction. Next, Post-Hoc Pairwise T tests were performed to identify the DEGs between Groups A and B, and *p*-Values were adjusted using the Bonferroni correction. LogFC was calculated using the formula (LogFC): log (A/B) = log (A) − log (B). Genes with an FDR-corrected *p*-Value < 0.1, a Bonferroni-corrected *p*-Value < 0.05, and a |LogFC| value ≥ 0.7 were considered differentially expressed in Group A compared to Group B. The R script used to perform the analysis is available at (https://github.com/nancyalnassar/DMD-gene-in-cancer/blob/main/Differential%20gene%20expression%20in%20primary%20tumor%20samples, accessed on 15 December 2022). Similarly, the differentially expressed miRNAs were identified.

### 2.8. Identification of DEGs in Tumor Cell Lines with Low and High Levels of DMD Gene Expression

We performed two comparisons in tumor cell lines from two distinct categories, carcinoma and sarcoma. A two-class comparison analysis was conducted using the DepMap custom analysis tool (https://depmap.org/portal/interactive/, accessed on 20 July 2022). The “Expression 21Q2 Public” was selected as the input dataset, and the cell lines with low *DMD* expression were used as the “in” group cell lines, while cell lines with high *DMD* expression were used as the “out” group cell lines. The analysis compared gene expression between the selected groups and generated estimates of effect size and corresponding q-values. The DEGs in each of these two comparisons (q-value < 0.05) were identified.

### 2.9. Identification of DEGs in Skeletal Muscle Samples from DMD Patients in Comparison to Healthy Skeletal Muscle

The GEO2R tool at the NCBI website (http://www.ncbi.nlm.nih.gov/geo/geo2r/, accessed on 23 May 2022) was used to identify the DEGs between 12 skeletal muscle samples collected from DMD patients and 11 samples collected from healthy donors. The default parameters of the GEO2R tool were used, and the NCBI-generated annotations were used to display the list of DEGs. The *p*-Values were adjusted using the FDR correction. Genes with an FDR-adjusted *p*-Value < 0.1 and a |LogFC| value ≥ 0.7 were considered differentially expressed between DMD and normal skeletal muscle samples. The R script generated by the GEO2R tool is available at (https://github.com/nancyalnassar/DMD-gene-in-cancer/blob/main/GEO2R%20Script, accessed on 15 December 2022).

### 2.10. Identification of DEGs between Hematological Malignancies Samples with Low and High Levels of Dp71 Expression

We compared gene expression in hematological malignancies samples with low vs. high levels of Dp71 expression from two TCGA and two TARGET studies. Samples (n = 286) were ranked based on the sum expression level of Dp71, Dp71b, and Dp71ab. Samples were then divided into three groups: Group A (samples at the bottom 33.3% of Dp71 expression), Group B (samples at the top 33.3% of Dp71 expression) and Group C (the remaining samples with medium level of Dp71 expression). Using the software R v4.0.4, an ANOVA was preformed to identify the DEGs between the three groups, and *p*-Values were adjusted using the FDR correction. Next, Post-Hoc Pairwise T tests were performed to identify the DEGs between Groups A and B, and *p*-Values were adjusted using the Bonferroni correction. Genes with an FDR-corrected *p*-Value < 0.1, a Bonferroni-corrected *p*-Value < 0.05, and a |LogFC| value ≥ 0.7 were considered differentially expressed in Group A compared to Group B.

### 2.11. Functional Enrichment Analysis

The EnrichR tool [44] was used to identify the Kyoto Encyclopedia of Genes and Genomes (KEGG) pathways and Gene Ontology (GO) Biological Process terms enriched in the DEGs in the previously mentioned comparisons. In comparisons where the number of DEGs was higher than 1000, only the top 1000 genes with the largest |LogFC| were used in the pathway and GO term enrichment analysis.

### 2.12. Survival Analysis

Samples (n = 6931) from 15 different TCGA primary tumors were ranked based on their levels of *DMD* gene expression. Next, the following survival endpoints were compared between patients at the bottom 25% of *DMD* expression and those at the top 25% of *DMD* expression: overall survival (OS), progression-free interval (PFI), disease-specific survival (DSS), and disease-free interval (DFI). 

Samples (n = 271) from two TCGA and two TARGET studies of hematological malignancies were ranked based on the sum expression level of Dp71, Dp71b, and Dp71ab. Next, OS was compared between patients at the bottom 25% of Dp71 expression and those at the top 25% of Dp71 expression. Kaplan-Meier curves were generated in GraphPad and analyzed using the log-rank test.

## 3. Results

### 3.1. Significant Expression of Dystrophin Transcript and Protein in a Range of Healthy Tissues

Expression data for 17 healthy tissues as well as skeletal muscle tissue from the GTEx database was examined. These were adrenal glands, bladder, breast, cervix, colon, esophagus, kidney, liver, lung, ovary, pancreas, prostate, skin, stomach, thyroid, uterus, and whole blood. *DMD* expression levels across this spectrum of healthy tissues were compared with the expression levels of two housekeeping genes (HKGs), *PKG1* and *HMBS*, which were identified as HKGs across 32 tissues in the GTEx database [36].

*DMD* expression averaged 79.3% of *PKG1* (range between 30.8 and 94.6%) and 114.7% of *HMBS* (43.7 to 137.4%) expression levels with the lowest expression found in whole blood (Appendix A). In skeletal muscle, *DMD* expression was 94.6% of *PKG1* and 138.2% of *HMBS*.

*DMD* expression relative to *PKG1* and *HMBS* was compared between skeletal muscle tissue and the previously mentioned 17 healthy tissues using a Kruskal-Wallis and Dunn’s multiple comparisons test. The expression of *DMD* relative to *PKG1* in skeletal muscle was comparable to that in the pancreas, uterus, and bladder (adjusted *p*-Value > 0.9999), and *DMD* expression relative to *HMBS* was comparable to that in the uterus, ovary, and bladder (adjusted *p*-Value > 0.9999).

Given this widespread presence of considerable *DMD* transcript levels, we investigated whether it is accompanied by protein expression. We interrogated mass spectrometry (MS) protein expression datasets available at Proteomics DB. MS identified the full-length dystrophin protein in a variety of normal tissues (Appendix A). Moreover, quantitative profiling of the proteome of 201 samples from 32 tissues in the GTEx database identified dystrophin as a housekeeping protein, as it was present in all of the 32 tissues analyzed [36]. Dystrophin protein expression was statistically significantly and positively correlated with *DMD* gene expression in those samples (Spearman correlation = 0.67, BH-adjusted *p*-Value < 0.1).

Finally, in the Protein Abundance Database (PAXdb), dystrophin expression was ranked in the top 25% of MS-quantified proteins in a range of tissues, such as fallopian tubes, esophagus, uterus, bladder, colon, prostate, and rectum, in addition to the heart. Thus, data from three databases demonstrated significant expression of dystrophin protein in a range of healthy tissues. Therefore, we investigated whether alterations in *DMD* expression might occur in tumors that originate from tissues not commonly associated with dystrophin protein expression or function.

### 3.2. Downregulation of DMD Gene Expression across Malignancies

We investigated RNA-seq expression data for 25 different types of primary tumors from the TCGA database (carcinomas, melanoma, lymphoma, and leukemia) and their corresponding NATs from the TCGA or healthy GTEx tissues. The analyzed primary tumors were acute myeloid leukemia (LAML), adrenocortical carcinoma (ACC), bladder urothelial carcinoma (BLCA), breast invasive carcinoma (BRCA), cervical and endocervical cancer (CESC), cholangiocarcinoma (CHOL), colon adenocarcinoma (COAD), diffuse large B-cell lymphoma (DLBC), esophageal carcinoma (ESCA), head and neck squamous cell carcinoma (HNSC), kidney chromophobe cell carcinoma (KICH), kidney clear cell carcinoma (KIRC), kidney papillary cell carcinoma (KIRP), liver hepatocellular carcinoma (LIHC), lung adenocarcinoma (LUAD), lung squamous cell carcinoma (LUSC), ovarian serous cystadenocarcinoma (OV), pancreatic adenocarcinoma (PAAD), prostate adenocarcinoma (PRAD), rectal adenocarcinoma (READ), skin cutaneous melanoma (SKCM), stomach adenocarcinoma (STAD), thyroid carcinoma (THCA), uterine carcinosarcoma (UCS), and uterine corpus endometrioid carcinoma (UCEC).

Global *DMD* gene expression was reduced in 20 out of 25 primary tumors in comparison to their corresponding control tissues (Figure 1; Appendix A). The largest difference in expression was found in primary breast invasive carcinoma (LogFC = −3.7), and the smallest in primary kidney papillary cell carcinoma (LogFC = −1). *DMD* gene expression was increased in primary tumors in two out of 25 comparisons, namely in primary thyroid carcinoma and diffuse large B-cell lymphoma, with LogFC values of 0.9 and 4.8, respectively. No significant expression changes were found in three comparisons (acute myeloid leukemia, kidney clear cell, and chromophobe cell carcinomas).

To examine the impact of batch effects resulting from differences in processing between TCGA and GTEx samples [45] in 13 comparisons, where *DMD* expression was compared between TCGA tumors and their corresponding NATs, we also compared *DMD* expression between these tumors and corresponding healthy tissues from the GTEx database. We found consistent results in 11 out of 13 comparisons, including BRCA, BLCA, UCEC, COAD, STAD, PRAD, LUSC, LIHC, LUAD, KICH, and THCA (Appendix A). In KIRP, *DMD* expression was reduced compared to kidney NAT in TCGA datasets (adjusted *p*-Value = 9.30 × 10^−22^), however, no statistically significant difference was found between *DMD* expression in KIRP and GTEx healthy kidney tissue samples (adjusted *p*-Value = 1). Moreover, there was no statistically significant difference in *DMD* expression between KIRC and paired kidney NAT samples (adjusted *p*-Value = 1), while *DMD* expression was found upregulated in KIRC samples compared to GTEx healthy kidney tissue (adjusted *p*-Value = 2.77 × 10^−8^). These inconsistencies might result from small sample sizes leading to a reduction in statistical power.

Given the well-known multiplicity of transcripts originating from the *DMD* gene and the variability in their expression patterns, a hierarchical clustering analysis was performed to identify changes in *DMD* expression profiles at the transcript level across various control tissues and corresponding tumors.

### 3.3. Hierarchical Clustering Analysis of the Relative Expression of DMD Transcripts Distinguishes Tumor Tissues

Of the top ten highly expressed transcripts (Appendix A), nine were predicted to be protein-coding (Appendix A). Three of these mRNAs were not canonical *DMD* gene transcripts. Of those, ENST00000358062.6 encoding H0Y304 protein is a poorly annotated transcript (Ensembl). The start of its CDS is unknown due to a 5′ truncation of the available sequence. The 5′ sequence upstream of the first exon (equivalent to exon 48 of the full-length transcript) is composed of the last 50 base pairs of the intron located between exons 47 and 48. Therefore, it is not clear whether this is a pre-mRNA sequence undergoing co-transcriptional splicing or a mature mRNA with this part of the intron spliced in. If the latter, the predicted protein encoded by such a transcript would have an N-terminus longer than Dp140 with valine as the putative initiation codon (UniProt). The transcript encoding H0Y864 appears to be a partial four-exon transcript, whose predicted protein sequence does not encode any functional domains (Appendix A). ENST00000475732.1 is a two-exon sequence not predicted to encode a protein.

Alternative splicing is a discernible feature of *DMD* transcripts found in the analyzed samples, with the splice variant Dp71b lacking exon 78 having the highest mean abundance level of all *DMD* transcripts across the analyzed tumor and control tissues, followed by Dp427m. The mRNA encoding Dp140c differs from the canonical Dp140 transcript, as it lacks exons 28 to 31 (equivalent to exons 71 to 74 of the full-length transcript) (Appendix A).

Hierarchical clustering analysis of these ten highly expressed transcripts in tumor and control tissues yielded six clusters. Appendix A reports other dendrogram cut-offs. We focused mainly on three of these clusters (Figure 2) that contained the majority of tumor and control tissues. The first cluster was composed of nine tumors and healthy whole blood and pancreas tissue. The second cluster was composed of 16 control tissues (including six healthy GTEx tissues and 10 NATs from the TCGA) as well as SKCM and PRAD tumor samples. Finally, the third cluster contained 12 tumors in addition to kidney and thyroid NATs from the TCGA.

The relative expression of the transcript encoding Dp427m was significantly lower in the first and third clusters, which were composed mainly of tumors, compared to the second cluster with a majority of control tissues, while the relative expression of Dp71 variants was higher in the first and third clusters (Appendix A). Specific transcriptomic alterations associated with decreased *DMD* gene expression are described in Section 3.7. Transcripts ENST00000493412.1 and ENST00000475732.1 did not show significantly different levels of expression between any of the clusters (Appendix A), and the differential expression of ENST00000358062.6 was not followed due to its uncertain annotation.

### 3.4. Changes in the Expression of Dp427m and Dp71 Transcripts across Malignancies

Given the importance of the full-length dystrophin, whose loss is responsible for Duchenne muscular dystrophy, and the predominance of the transcript encoding Dp427m (Appendix A), its expression patterns were analyzed in more detail. As Dp71 is the isoform most widely expressed across the body and its splice variants were among the top highly expressed *DMD* transcripts in tumor and control tissues (Appendix A), we also compared expression patterns of this transcript (Appendix A). Dp427m expression was decreased in primary tumors compared to control tissues in 17 out of 25 comparisons with LogFC values ranging from −7.2 to −2.7 in primary uterine carcinosarcoma and primary thyroid carcinoma, respectively (Figure 3). There was a statistically significant change in the expression of transcripts encoding Dp71 in 10 out of 25, Dp71ab in 13 out of 25, while Dp71b was altered in 20 out of 25 comparisons (a decrease in 17 and an increase in three) (Appendix A).

Further analysis of these *DMD* transcripts showed that changes in overall *DMD* expression levels in two tumor categories were confounded by the opposing dysregulation of Dp427m and Dp71 variants. Specifically, while *DMD* expression was higher in primary thyroid carcinoma compared to thyroid NAT, Dp427m expression there was lower compared to thyroid NAT, and the increase in overall *DMD* gene expression resulted from the elevated expression of Dp71b (Appendix A). In contrast, in primary pancreatic adenocarcinoma, total *DMD* expression was lower compared to the healthy pancreas tissue, but this decrease resulted from the lower expression of transcripts encoding Dp71 variants, while Dp427m expression was higher in this tumor type (LogFC = 3.7 compared to healthy pancreas tissue) (Appendix A).

### 3.5. DMD Expression Downregulation Occurs Irrespective of Somatic Mutations within the DMD Locus

Next, using datasets for samples from 23 of the previously mentioned TCGA tumors (LAML and SKCM samples were excluded as they did not have mutation and SCNA details available) we investigated the association between *DMD* expression and mutations in coding (CRs) and non-coding regions (NCRs) as well as SCNAs in the *DMD* gene.

The majority of tumor samples had no identified CR mutations in the *DMD* gene locus (6201 out of 6751). A univariate GLM analysis was carried out to assess the effect of CR mutations and gender (to account for the X-chromosome localization of the *DMD* gene) as well as their interaction on *DMD* expression. The GLM indicated a significant effect for CR mutations on *DMD* expression (*p* = 0.007). There was no effect for gender (*p* = 0.703) or the interaction between CR mutations and gender (*p* = 0.648). Samples with missense and multiple mutations had significantly lower levels of *DMD* expression compared to samples with no CR mutations in the *DMD* locus (*p* < 0.001 and *p* = 0.043, respectively) (Figure 4A). A Chi Square test revealed that there is an overrepresentation of *DMD* mutations in tumor samples with low *DMD* expression (X^2^(1) = 45.44, *p* < 0.0001). Specifically, 11.7% of samples in the low *DMD* group (237 out of 2025) had *DMD* CR mutations compared to 5.73% (116 out of 2025) in the high *DMD* group. However, in the low *DMD* group, 88.3% of samples (1788 out of 2025) did not have any detectable mutations in the CRs of the *DMD* locus.

As for NCR mutations, 98.2% of tumor samples (6632 out 6751) had no mutations in the non-coding regions of the *DMD* gene. NCR mutations were found to have a significant effect on *DMD* expression (*p* = 0.031), but no effect was found for gender (*p* = 0.104) or the interaction between gender and NCR mutations (*p* = 0.698). Samples with intronic mutations had significantly lower *DMD* expression compared to samples without any *DMD* NCR mutations (*p* = 0.02) (Figure 4B). A Chi Square test revealed that there is an overrepresentation of *DMD* NCR mutations in tumor samples with low *DMD* expression (X^2^(1) = 18.71, *p* < 0.0001). In the low *DMD* group, 3.01% of tumor samples (61 out of 2025) had *DMD* NCR mutations compared to 1.09% of samples (22 out of 2025) in the high *DMD* group. However, 96.99% of samples in the low *DMD* group (1964 out of 2025) did not have any NCR mutations in the *DMD* locus.

Regarding SCNAs, 66.7% of samples had a normal copy number of the *DMD* locus (4503 out of 6751). Both SCNAs (*p* < 0.001) and gender (*p* = 0.006) had a significant effect on *DMD* expression. However, the interaction between the two factors did not have a significant effect. Samples with a normal *DMD* copy number had higher expression compared to samples with deep and shallow deletions and those with gains (*p* < 0.001). Additionally, samples with deep deletions had lower expression compared to those with amplifications (*p* = 0.002) (Figure 4C).

In samples from female patients, we found that there is an overrepresentation of *DMD* SCNAs in the low *DMD* group (X^2^(1) = 36.40, *p* < 0.0001), where 39.08% of samples (399 out of 1021) had SCNAs in the *DMD* locus compared to 26.54% in the high *DMD* group (271 out of 1021). In the low *DMD* group, 60.92% of samples (622 out of 1021) did not have any SCNAs.

In samples from male patients, we also found that there is an overrepresentation of *DMD* SCNAs in the low *DMD* group (X^2^(1) = 69.82, *p* < 0.0001), where 44.18% of samples (444 out of 1005) had SCNAs in the *DMD* locus compared to 26.37% in the high *DMD* group (265 out of 1005). In the low *DMD* group, 55.82% of samples had no SCNAs.

We further investigated the association between *DMD* expression and mutations using data from 921 tumor cell lines (cBioPortal). The majority of tumor cell lines had no identified mutations within the *DMD* gene region (n = 773), while the remaining had missense (n = 120), truncating (n = 11), splice (n = 4), and multiple mutations (n = 13). We found that there is an overrepresentation of *DMD* mutations in tumor cell lines with low levels of *DMD* expression (X^2^(1) = 9.727, *p* = 0.0018). However, only 23% of tumor cell lines with low *DMD* gene expression (63 out of 276) had *DMD* mutations, but the majority of cell lines (213 out of 276) had low levels of *DMD* expression without any detectable mutations.

Therefore, downregulation of *DMD* expression in tumors cannot be simply attributed to somatic mutations or copy number alterations within the *DMD* locus, but rather involves a regulatory mechanism.

### 3.6. Association between DMD Expression and Cancer Stage and Patient’s Age

We investigated the association between *DMD* expression and stage in 18 different types of primary cancers with available data for stage, gender, and patient’s age at the initial diagnosis. These tumors included ACC, BLCA, BRCA, CHOL, COAD, ESCA, HNSC, KICH, KIRC, KIRP, LIHC, LUAD, LUSC, PAAD, READ, SKCM, STAD, and THCA. The ages of patients ranged from 14 to 90 years old.

A GLM analysis was performed to assess the effect of cancer stage on *DMD* expression while accounting for gender and age differences. Cancer stage was found to have an effect on *DMD* expression (*p* = 0.035). While no effect was identified for gender differences (*p* = 0.735), the interaction between cancer stage and gender was found to have an impact on *DMD* expression (*p* = 0.010). Age of patients was also found to have an impact on *DMD* gene expression (*p* < 0.001) as well as the interaction between age and cancer stage (*p* = 0.003).

Samples from patients with stage I cancer had significantly higher levels of *DMD* gene expression compared to those with stage II (LogFC = 0.72, *p* < 0.001), stage III (LogFC = 0.50, *p* < 0.001), and stage IV cancer (LogFC = 0.69, *p* < 0.001) (Figure 5A). Moreover, samples from younger patients had significantly higher *DMD* expression compared to samples from older patients (Figure 5B).

Twelve of the tumor tissues used in the previous analysis have corresponding healthy tissues in the GTEx database: adrenal glands, bladder, breast, colon, esophagus, kidney, liver, lung, pancreas, skin, stomach, and thyroid. We examined whether there is an association between *DMD* gene expression and age in these healthy tissue samples (n = 2863). The ages of donors ranged from 20 to 79 years old, and samples were grouped into six age groups: 20–29, 30–39, 40–49, 50–59, 60–69, and 70–79 years. There was no statistically significant effect for age (*p* = 0.484), gender (*p* = 0.647), or their interaction (*p* = 0.591) on *DMD* expression in these healthy samples.

### 3.7. Decreased DMD Gene Expression in Primary Tumors Is Associated with Specific Transcriptomic Alterations 

Expression of the *DMD* gene is significantly altered in tumors, with the majority having lower expression levels compared to their respective control tissues. This downregulation predominantly affects the transcript encoding the full-length dystrophin due to a regulatory alteration. To identify downstream molecular pathways that could be associated with such *DMD* downregulation (analogous to the impact of full-length dystrophin loss in muscle), we compared transcriptomes of primary tumor samples with low vs. high *DMD* gene expression from 15 different tumor types: BRCA, BLCA, UCEC, CESC, OV, COAD, STAD, PRAD, LUSC, ESCA, HNSC, LUAD, KIRP, and THCA, where Dp427m was found to be downregulated compared to control tissues. In addition, we included sarcoma (SARC) in this comparison because sarcomas originate from tissues (muscle and bone) known to express the *DMD* gene. For clarification, RNA-seq data for control tissues for sarcomas was not available in the TCGA TARGET GTEx cohort, and therefore *DMD* gene expression in sarcomas vs. control tissues could not be compared. In each tumor type, the DEGs between samples at the bottom 33.3% and top 33.3% of *DMD* expression were identified and used to perform a pathway enrichment analysis. Figure 6 shows the combined score values (−Log (*p*-Value) × odds ratio) for KEGG pathways that were found to be enriched in DEGs in more than 50% of the analyzed primary tumors (adjusted *p*-Value < 0.05). 

The differentially expressed transcripts in tumors with low *DMD* expression showed enrichment in the ECM-receptor interaction pathway in all 15 primary tumors analyzed. Calcium signaling and protein digestion and absorption were enriched in 13 tumors, cell adhesion molecules in 12, focal adhesion in 11, PI3K and cAMP signaling in 10, Wnt signaling in 9, and cGMP-PKG signaling and axon guidance were enriched in 8 comparisons.

The specific GO Biological Process terms enriched in DEGs in primary tumors with low vs. high *DMD* expression were also identified (Figure 7). The GO terms enriched in more than 50% of tumors were: extracellular matrix organization (14 out of 15 tumors), axonogenesis (12), regulation of cell migration, synapse organization and nervous system development (11), skeletal system development, regulation of ERK1 and ERK2 cascade (10), and calcium ion transmembrane import into cytosol (8). Lists of DEGs in each comparison and results of the pathway and GO term analysis can be found in Appendix A.

In order to confirm that these transcriptomic changes between primary tumor samples with low vs. high *DMD* expression are associated with *DMD* downregulation and not a result of other factors, we repeated the analysis for each tumor type using three groups of tumor samples identical in size to the groups used in the previous analysis, but that were chosen at random. *DMD* gene expression was not statistically significantly different between the three random groups in all of the 15 tumors, and no statistically significant differences were found in the transcriptomes between these groups, confirming the specific gene expression alterations to be associated with *DMD* downregulation (Appendix A).

### 3.8. Transcriptomic Alterations Associated with Decreased DMD Gene Expression in Tumor Cell Lines

To further confirm that these transcriptomic changes are evoked by *DMD* downregulation specifically in cancer cells rather than originating from *DMD* expression in the stromal or infiltrating immune cells present in tumor tissue samples, we conducted a two-class comparison analysis using the DepMap portal to identify DEGs between cell lines of the same tumor origin (carcinoma and sarcoma) but with low vs. high level of *DMD* gene expression. While the *DMD* gene was not found to be essential for tumor cell line survival [46,47], its altered expression may have an important common effect across malignancies.

One-hundred and forty tumor cell lines were grouped into four groups based on their level of *DMD* expression (low or high) and their origin (carcinoma or sarcoma) (Appendix A). The ranges for *DMD* expression for these cell lines were as follows (Unit: log_2_ (TPM + 1)): carcinoma cell lines with high (3.3–7.4) and low (0–0.01) *DMD* expression, sarcoma cell lines with high (4.07–7.6) and low (0–0.06) *DMD* expression. The carcinoma cell lines used in this analysis originated from tissues where primary tumors were found to have a lower level of Dp427m transcript compared to control tissues. The two-class comparison analysis between cell lines with low vs. high *DMD* expression identified 998 DEGs for carcinoma, and 543 DEGs for sarcoma cell lines. Interestingly, in carcinoma cell lines with low *DMD* expression, the majority of DEGs were downregulated (n = 976 out of 998). Figure 6 shows the combined score values (−Log (*p*-Value) × odds ratio) for KEGG pathways that were found to be enriched in DEGs in these two comparisons (adjusted *p*-Value < 0.05).

Pathway enrichment for the DEGs in carcinoma cell lines with low *DMD* expression suggested that *DMD* downregulation may affect the following KEGG pathways: ECM-receptor interaction (*p* = 0.037), protein digestion and absorption (*p* = 0.015), focal adhesion (*p* = 0.029), PI3K-Akt signaling (*p* = 0.021), cAMP signaling (*p* = 0.044), cGMP-PKG signaling (*p* = 0.024), and axon guidance (*p* = 0.002). Notably, these pathways were also enriched in the DEGs in more than 50% of primary tumors with low *DMD* expression.

Similar to carcinoma cell lines, the following KEGG pathways were enriched in the DEGs in sarcoma cell lines with low *DMD* expression: ECM-receptor interaction (*p* = 0.048), PI3K-Akt signaling (*p* = 0.041), and cAMP signaling (*p* = 0.034). The calcium signaling pathway, which was enriched in the DEGs in more than 50% of primary tumors with low *DMD* expression, was also enriched in the DEGs in the comparison of sarcoma cell lines (*p* = 7.24 × 10^−4^).

Moreover, GO Biological Process term analysis for DEGs in carcinoma cell lines with low vs. high *DMD* expression (Figure 7 and Appendix A) identified the following terms: extracellular matrix organization (*p* = 5.22 × 10^−4^), cell junction assembly (*p* = 0.012), and positive regulation of epithelial to mesenchymal transition (*p* = 0.038). Terms related to the development of the CNS and synaptic transmission: nervous system development (*p* = 8.39 × 10^−5^), neuron migration (*p* = 8.39 × 10^−5^), and regulation of neuron projection development (*p* = 1.04 × 10^−5^) were also found.

The GO Biological Process term regulation of cell migration was enriched (*p* = 0.02) in DEGs in sarcoma cell lines with low vs. high *DMD* expression (Figure 7 and Appendix A). Lists of the DEGs in each comparison and results of the pathway and GO term analysis can be found in Appendix A.

### 3.9. Transcriptomic Alterations in Duchenne Skeletal Muscle Compared to Healthy Muscle Samples

Next, we investigated whether the transcriptomic changes resulting from *DMD* downregulation in primary tumors and tumor cell lines are similar to those observed in skeletal muscles of DMD patients. We compared gene expression data from 12 DMD skeletal muscle and 11 healthy muscle samples. The GEO2R tool identified 1160 genes to be differentially expressed between DMD and healthy skeletal muscle. Figure 6 shows the combined score values (−Log (*p*-Value) × odds ratio) for the KEGG pathways that were found to be enriched in the top 1000 DEGs with the highest |LogFC| values in this comparison (adjusted *p*-Value < 0.05).

Pathway enrichment analysis indicated *DMD* downregulation to be associated with changes in the following KEGG pathways: ECM receptor interaction (*p* = 7.82 × 10^−11^), calcium signaling pathway (*p* = 8.80 × 10^−8^), protein digestion and absorption (*p* = 3.67 × 10^−8^), cell adhesion molecules (*p* = 1.55 × 10^−4^), focal adhesion (*p* = 2.87 × 10^−12^), PI3K-Akt signaling (*p* = 7.82 × 10^−11^), cAMP signaling (*p* = 0.020), cGMP-PKG signaling (*p* = 0.007), and axon guidance pathways (*p* = 0.030). These very pathways were enriched in the DEGs in more than 50% of primary tumors with low *DMD* expression, and also enriched in DEGs in carcinoma and sarcoma cell lines with low vs. high *DMD* expression.

GO Biological Process term analysis for DEGs in DMD skeletal muscle compared to healthy muscle (Figure 7) identified the following terms: extracellular matrix organization (*p* = 6.25 × 10^−17^), axonogenesis (*p* = 9.35 × 10^−5^), regulation of cell migration (*p* = 3.39 × 10^−12^), synapse organization (*p* = 0.009), nervous system development (*p* = 1.87 × 10^−4^), skeletal system development (*p* = 2.22 × 10^−4^), regulation of ERK1 and ERK2 cascade (*p* = 5.15 × 10^−8^), and calcium ion transmembrane import into cytosol (*p* = 0.001). These GO terms were enriched in DEGs in more than 50% of primary tumors with low *DMD* expression. GO terms that were found to be in common with carcinoma cell lines with low *DMD* expression were: extracellular matrix organization, nervous system development, cell junction assembly (*p* = 0.004), regulation of neuron projection development (*p* = 0.005), neuron migration (*p* = 0.004), and positive regulation of epithelial to mesenchymal transition (*p* = 2.52 × 10^−4^). Finally, the GO term, regulation of cell migration, was found to be in common with sarcoma cell lines with low *DMD* expression.

Furthermore, functional enrichment tests of the dystrophin protein-protein interaction network constructed using high confidence PPI information demonstrated that the enriched functional terms were consistent with the pathways found to be significantly enriched in the aforementioned comparisons of primary tumors, tumor cell lines, and DMD skeletal muscle: extracellular matrix organization (*p* = 1.10 × 10^−18^), axon guidance (*p* = 9.45 × 10^−10^), focal adhesion (*p* = 3.27 × 10^−14^), and PI3K-Akt signaling pathways (*p* = 9.27 × 10^−9^) (Appendix A).

### 3.10. Low DMD Gene Expression Is Associated with Poor Survival in Patients with 15 Different Primary Tumor Types

Given these similarities between pathways altered in tumors and in the lethal muscle disease, we examined the association between *DMD* gene expression and patients’ survival in the following tumor types: BRCA, BLCA, UCEC, CESC, OV, COAD, STAD, PRAD, LUSC, ESCA, HNSC, LUAD, KIRP, THCA, and SARC. We compared overall survival (OS), progression-free interval (PFI), disease-specific survival (DSS), and disease-free interval (DFI) endpoints between patients at the bottom 25% of *DMD* expression and those at the top 25% across all the aforementioned tumor types. OS was lower in the low *DMD* group (HR 1.33; 95% 1.17, 1.51; *p* < 0.0001) with 2417 days compared to 3253 for the high *DMD* group. PFI was also decreased in the low *DMD* group (HR 1.28; 95% 1.14, 1.45; *p* < 0.0001) with 2472 days compared to 3669 days, respectively. Finally, the low *DMD* group had lower DSS (HR 1.46; 95% 1.24, 1.72; *p* < 0.0001) and DFI (HR 1.30; 95% 1.06, 1.59; *p* = 0.012) compared to the high *DMD* group (Figure 8A).

In order to confirm that these changes in survival outcomes between patients with low vs. high *DMD* expression in their tumors are associated with *DMD* downregulation and not a result of other factors, we repeated the survival analysis using two groups of patients chosen at random, and no statistically significant differences in survival endpoints were found between the two.

### 3.11. Transcriptomic Alterations in Hematological Malignancies with Low vs. High Dp71 Expression

The hierarchical clustering analysis revealed that the blood malignancies acute myeloid leukemia (LAML) and diffuse large B-cell lymphoma (DLBC) had a unique pattern of *DMD* transcripts, to the point that these two malignancies were classified as a separate cluster (Figure 2). While no changes in Dp427m expression were observed in both TCGA datasets for LAML and DLBC compared to healthy blood, Dp71 levels including its splice variants Dp71b and Dp71ab were higher in these tumors (Figure 3, Appendix A). Levels of Dp71 and its splice variants were also found higher in two TARGET datasets (acute myeloid leukemia and acute lymphoblastic leukemia) compared to GTEx healthy whole blood (Appendix A). However, we note that age differences between TARGET and GTEx donors might be a confounding factor when interpreting these results.

Next, we compared gene expression between samples from the previously mentioned TCGA and TARGET datasets at the bottom 33.3% and top 33.3% of Dp71 expression across all tumor types. The pathways enriched in the top 1000 DEGs with the largest |LogFC| were: protein digestion and absorption (*p* = 8.59 × 10^−5^), ECM-receptor interaction (*p* = 3.87 × 10^−4^), focal adhesion (*p* = 0.003), and PI3K-Akt signaling pathway (*p* = 0.02). The identified GO Biological Process terms for these genes were extracellular matrix organization (*p* = 1.57 × 10^−10^), regulation of angiogenesis (*p* = 7.94 × 10^−4^), skeletal muscle development (*p* = 8.38 × 10^−4^), positive regulation of MAPK cascade (*p* = 0.005), regulation of ERK1 and ERK2 cascade (*p* = 0.006), regulation of cell migration (*p* = 0.01), and positive regulation of calcium ion import (*p* = 0.02) (Appendix A). Thus, these pathways and processes were identical with those found in comparisons with low vs. high expression of Dp427m in completely unrelated malignancies. A comparison between random groups of samples from these hematological tumors did not reveal any statistically significant changes in gene expression.

### 3.12. Low Expression of Dp71 Is Associated with Poor Survival in Patients with Hematological Malignancies

We compared OS between hematological malignancies patients with low and high expression of Dp71 and its splice variants. OS was lower in the low Dp71 group (HR 2.39; 95% 1.44, 3.99; *p* = 0.0003) with 792 days compared to 1992 for the high Dp71 group (Figure 8B). For specificity testing, the survival analysis was conducted using two random groups of patients with these hematological malignancies and no statistically significant difference in overall survival was found.

Thus, while specific dystrophins are differentially regulated across various tumors, the low expression of both full-length dystrophin and the Dp71 variants in the analyzed tumors is associated with analogous molecular alterations and significantly decreased patient survival.

## 4. Discussion

We found a significant downregulation of *DMD* gene expression across diverse primary tumors. Both full-length and truncated dystrophin variants were differentially expressed, and hierarchical clustering of the top highly expressed transcripts distinguished tumors from corresponding control tissue samples. A similar trend for *DMD* downregulation across carcinomas has been described previously [24], and our analysis discriminating specific *DMD* transcripts showed that levels of Dp427m mRNA were statistically significantly decreased in the clusters composed mainly of tumor tissues (Appendix A), suggesting a specific impact of the loss of the full-length transcript. Indeed, pancreatic adenocarcinoma was the only primary tumor with higher levels of Dp427m transcripts compared to healthy pancreas tissue. In contrast, the relative expression of Dp71 splice variants was increased in tumor clusters.

In contrast to the majority of primary tumor samples, in hematological malignancies, total *DMD* expression and the expression of the full-length and Dp71 transcripts was higher or unchanged when compared to normal blood, which also showed the lowest *DMD* expression relative to housekeeping genes of all the healthy tissues analyzed (Appendix A). However, human and mouse hematopoietic stem cells were found to express Dp71, and its expression was decreasing with cell differentiation (manuscript submitted), which agrees with whole blood showing the lowest *DMD* expression. Interestingly, while Dp71 expression is found upregulated in hematological malignancies, its low levels were found to be associated with the very same dystrophic molecular alterations in cancer cells.

Despite the advantage that using NATs as control samples in cancer studies reduces individual and anatomical site-specific confounding factors and eliminates technical interlaboratory differences, it was found that these tissues are distinct from healthy and tumor tissues and represent a unique intermediate state between them [48]. Although we showed that the results of comparing *DMD* expression between TCGA tumors and their corresponding NATs were consistent with those of comparing *DMD* expression between TCGA tumors and healthy GTEx tissues in 11 out of 13 comparisons, the unique transcriptomic profile of NATs might explain why thyroid NATs and kidney NATs from three different kidney TCGA tumors clustered with tumor tissues in the third cluster. The opposite was observed in the second cluster, where PRAD and SKCM samples clustered with control tissues. This might be the result of a high proportion of non-tumor cells in these tumor samples.

Given some evidence of a causative link between *DMD* downregulation and phenotypic changes in tumor cells [17,23] and that alterations in Duchenne, such as increased cell proliferation, abnormalities in adhesion, migration, and invasion [11,12] are commonly associated with malignancy, transcriptomes of primary tumor samples as well as tumor cell lines with low vs. high levels of *DMD* gene expression were compared.

While no causation can be confirmed at this stage, it is important to note that *DMD* dysregulation was associated with specific transcriptomic changes across 15 primary tumors and 140 various tumor cell lines (summarized in Figure 9). 

Functional enrichment analysis showed that the pathways and GO Biological Process terms significantly enriched in DEGs in primary tumors and cell lines with low vs. high *DMD* expression were consistent with the pathways and GO terms enriched in DEGs in DMD skeletal muscle as well as the functional dystrophin PPI network (Appendix A). Key pathways altered, including cell adhesion, ECM interactions, and PI3K-Akt signaling, correspond to alterations found in Duchenne patients’ cells [49,50]. The calcium signaling pathway enriched in DEGs in tumors samples with low *DMD* expression from 13 out of the 15 analyzed primary tumors, and well as in sarcoma cell lines with low *DMD* expression, agrees with the dysregulation of calcium signaling across a whole spectrum of dystrophic cells (reviewed in [51]), as do GO terms related to the regulation of the developmental mechanisms [8]. Thus, the *DMD* gene may play similar roles in cancer and development, two processes showing biological and molecular similarities [52].

The protein digestion and absorption amongst the top pathways identified is somewhat surprising, but it is also present in DMD muscles and must reflect the overlap between DEGs. For example, 17 genes in this pathway are shared with the ECM receptor interaction, focal adhesion, and PI3K-Akt, and six genes are shared with the cAMP and cGMP-PKG signaling pathways.

Dystrophin in tumor cell lines did not correlate strongly with the presence of its established DAPC partners, suggesting that it may have a different role(s) than those in muscle cells (Appendix A). This is unsurprising, given that DAPs are known to differ in different tissues, with muscle and brain being the most notable examples. But even within muscle, the dystrophin interactome changes with differentiation, and functionally distinct DAPs exist in satellite cells and myofibers.

The importance of the *DMD* gene in tumorigenesis is supported by the finding that low *DMD* expression was associated with poor survival outcomes in patients with 15 different types of tumors (14 carcinomas and sarcoma). The overall survival of cancer patients with decreased *DMD* expression in tumors was 27 months lower than that of patients with high *DMD* expression. However, since the low and high *DMD* groups used in this analysis were composed of 15 tumor types, the number of samples for each of those types is highly variable between the two groups, and this could possibly be a confounding factor when interpreting the results. In other studies, mutations in the *DMD* gene were associated with poor overall survival of patients of two out of 11 analyzed tumors, namely uterine corpus endometrioid carcinoma and breast invasive carcinoma [24]. Dystrophin protein was also identified as a survival biomarker in upper gastrointestinal cancer patients, as poorer survival was observed in patients with low compared to high levels of dystrophin protein [53].

In our analyses, the overall survival of patients suffering from hematological malignancies with decreased Dp71 expression was about 39 months lower than that of patients with high Dp71 expression. However, in low-grade glioma [54] and B-cell chronic lymphocytic leukemia [28], high Dp71 expression was previously associated with poor patient survival.

We found *DMD* expression to be associated with the tumor stage. Samples from patients with stage I had significantly higher levels of *DMD* expression compared to higher stages after controlling for age and gender differences.

*DMD* expression was also found to decrease with the age of onset, as samples from younger patients had higher *DMD* expression compared to samples from older patients. This association between *DMD* expression and age was unique to tumor tissues, as no such association was found by us in the corresponding healthy tissues from the GTEx database, and also in a meta-analysis that identified genes with age-associated expression in human peripheral blood samples [55].

It is worth noting that recent studies identified increased frequency of rhabdomyosarcomas in DMD patients [56,57] which agrees with previous data on spontaneous rhabdomyosarcomas in dystrophic mice [58].

Crucially, we demonstrated downregulation of *DMD* expression in tumors with incidence increasing with age. Therefore, with improved therapies, the risk of malignancy in DMD should be considered. As for hematological malignancies, which frequently affect children, poor survival was associated with Dp71 downregulation. Expression of this dystrophin is not affected in the vast majority of DMD patients.

A further indication of the functional significance of the *DMD* gene in tumors is that *DMD* downregulation across various malignancies involves regulatory changes, not just results from somatic gene mutations. Although, as expected, the presence of some types of somatic mutations and SCNAs was associated with lower levels of *DMD* expression, in about 88% of tumor samples and 77% of cell lines, *DMD* downregulation could not be linked to mutations in the coding regions of the *DMD* gene. Moreover, *DMD* downregulation was not a result of SCNAs in about 61% of samples from female patients and about 56% of samples from male patients. While deletions were described as a causative factor for the downregulation of *DMD* gene expression in some tumors [17,23,30], significantly reduced expression was also found in the absence of deletions or nonsense mutations [27], in agreement with our comprehensive analysis across different malignancies. Moreover, in primary pancreatic adenocarcinoma, Dp427m transcript was increased while Dp71 expression was reduced (Appendix A). Given the *DMD* gene structure, such an expression pattern can only be explained by differential regulation. Targeted degradation of dystrophin transcripts was suggested before [59], and recently, an epigenetic mechanism responsible for reduced *DMD* transcript levels has been described [60]. These data indicate that, rather than simply being an effect of random mutations, quite likely to occur in this large gene, *DMD* alterations in tumor cells may have a complex regulatory nature involving mechanisms such as transcriptional regulation, chromatin remodeling, or transcript degradation. Given that miRNAs might be responsible for the differential regulation of *DMD* expression in tumor samples, we investigated but did not find any differentially expressed miRNAs in tumor samples with low vs. high *DMD* expression that were common to all of the 15 primary tumor types analyzed (Appendix A).

Thus, specific alterations in *DMD* gene expression are a common feature across a spectrum of malignancies, including those originating from tissues previously not associated with the expression of the full-length dystrophin. Yet, we found *DMD* transcript levels in these tissues to be comparable to the levels of housekeeping genes. Moreover, interrogation of proteomics datasets demonstrated a much wider distribution profile for the full-length dystrophin protein (Appendix A). Interestingly, according to the Human Protein Atlas [38], the antibody HPA002725, directed against amino acids 186–333 of the full-length dystrophin, detected moderate cytoplasmic and/or membrane staining in a range of normal tissues in addition to the expected staining in skeletal and cardiac muscle and the CNS. However, the antibody HPA023885, raised against amino acids 2843–2992 and therefore supposed to detect all dystrophin isoforms upstream of Dp71, showed staining in skeletal and cardiac muscle, while other tissues were negative. This staining pattern disagrees with the established expression of Dp260, Dp140, and Dp116, which is broader than that of Dp427 isoforms, and so it could not be accurate. Given the high sensitivity and specificity of the mass-spec [61,62], identification of the full-length dystrophin in a wide spectrum of normal tissues using this latter method is likely to represent the true expression status. Thus, dystrophin may be present in many tissues at low levels and/or in a tightly controlled spatiotemporal manner, which might be missed using standard detection methods.

The molecular signature associated with decreased *DMD* expression in tumors and corresponding tumor cell lines is concordant with that found in Duchenne muscular dystrophy. The *DMD* gene encodes a spectrum of dystrophin isoforms, but significant expression of the majority of these appeared to be restricted to specific tissues. While loss of the full-length dystrophin results in Duchenne muscular dystrophy, mutations additionally disrupting other isoforms result in exacerbated phenotypes [36,63]. In skeletal myofibers, cardiomyocytes, and neurons, which express the highest levels of the full-length dystrophin, this protein has been described to serve as a structural scaffold for proteins engaged in ECM and cell-cell interactions, and in intracellular signaling. However, more recent data demonstrate that the loss of *DMD* expression impacts a broader spectrum of cells than those affected in DMD. In myoblasts [10,16,64,65], lymphocytes [33], endotheliocytes [32,66,67], mesodermal [8], and myogenic cells [11,15], loss of *DMD* expression leads to significant abnormalities. Moreover, the same abnormalities can occur in very distinct cells, e.g., calcium dys-homeostasis was found across multiple cells [51], and the damaging purinergic phenotype affects myoblasts and lymphocytes [33,65]. Some old findings, such as platelet abnormalities [68], have recently been vindicated [69]. Yet, these defects cannot be clearly attributed to the loss of interaction between dystrophin and the known dystrophin-associated proteins. Indeed, these cell-autonomous defects appear to affect dystrophic cells, which, when healthy, express the 14-kb *DMD* transcript, but were not shown to produce detectable levels of full-length dystrophins. This phenomenon, where expression of the 14-kb *DMD* transcript in cells such as myoblasts and lymphocytes does not correlate with detectable dystrophin protein, was known for decades but just disregarded as an “illegitimate transcription” [70].

Our data suggest that more attention should be given to the subtler *DMD* gene functions, beyond those causing the main symptoms of DMD. Such studies have a potential to identify new therapeutic targets for the treatment of this debilitating and still incurable disease. Moreover, given the poor survival rate of patients with tumors downregulating dystrophin, the *DMD* gene may be important in oncology.

## 5. Conclusions

The data presented here indicate that the current view on the role of the *DMD* gene in health and disease requires significant re-evaluation. The functional significance of the *DMD* gene is far more widespread than its identified roles in Duchenne MD, and its downregulation in tumors is of particular importance as it was found to be associated with specific transcriptomic changes and reduced survival of patients across a variety of malignancies.

## Figures and Tables

**Figure 1 cancers-15-01378-f001:**
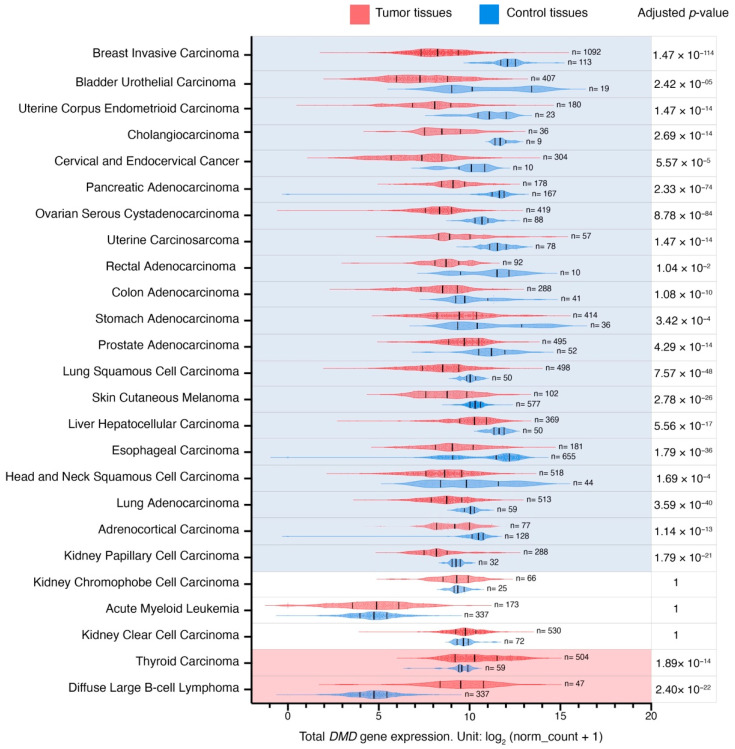
*DMD* gene expression level in primary tumors and corresponding control tissues. Red and blue violin plots represent tumor and control tissues, respectively. Vertical black lines represent the median and quartiles. Comparisons (n = 20) with a statistically significant downregulation of *DMD* expression in tumor samples compared to controls are highlighted in blue, and comparisons (n = 2) with a statistically significant upregulation of *DMD* expression in tumor samples compared to controls are highlighted in red (*p* < 0.05, Appendix A). The *p*-Values were calculated using the UCSC Xena Browser using a two-tailed Welch’s t test and adjusted for multiple testing using the Bonferroni correction.

**Figure 2 cancers-15-01378-f002:**
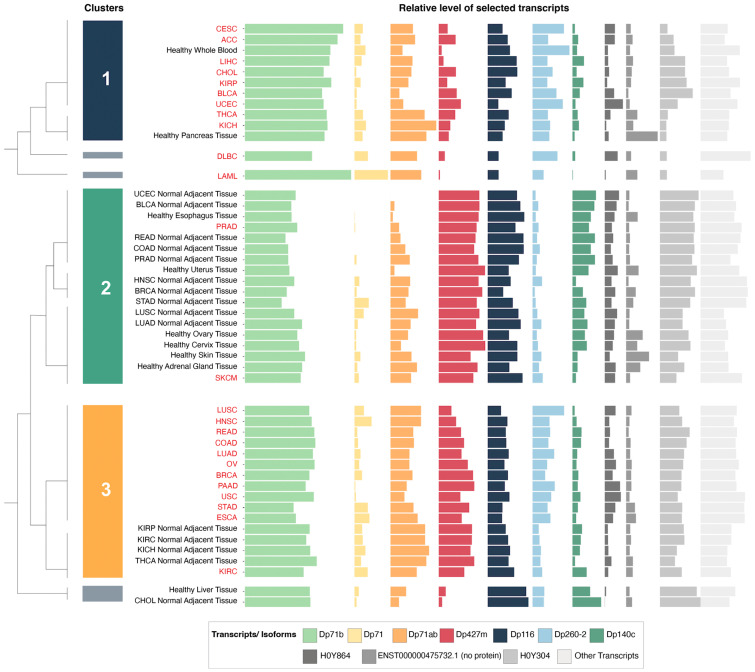
Bar plots representing the relative expression levels of the top 10 highly expressed *DMD* transcripts in tumor (text in red) and control tissues (text in black) and a dendrogram displaying the clusters identified by the hierarchical clustering analysis. Abbreviations: CESC (cervical and endocervical cancer), ACC (adrenocortical carcinoma), LIHC (liver hepatocellular carcinoma), CHOL (cholangiocarcinoma), KIRP (kidney papillary cell carcinoma), BLCA (bladder urothelial carcinoma), UCEC (uterine corpus endometrioid carcinoma), THCA (thyroid carcinoma), KICH (kidney chromophobe cell carcinoma), DLBC (diffuse large B-cell lymphoma), LAML (acute myeloid leukemia), PRAD (prostate adenocarcinoma), HNSC (head and neck squamous cell carcinoma), LUSC (lung squamous cell carcinoma), LUAD (lung adenocarcinoma), SKCM (skin cutaneous melanoma), READ (rectal adenocarcinoma), COAD (colon adenocarcinoma), OV (ovarian serous cystadenocarcinoma), BRCA (breast invasive carcinoma), PAAD (pancreatic adenocarcinoma), USC (uterine carcinosarcoma), STAD (stomach adenocarcinoma), ESCA (esophageal carcinoma), KIRC (kidney clear cell carcinoma).

**Figure 3 cancers-15-01378-f003:**
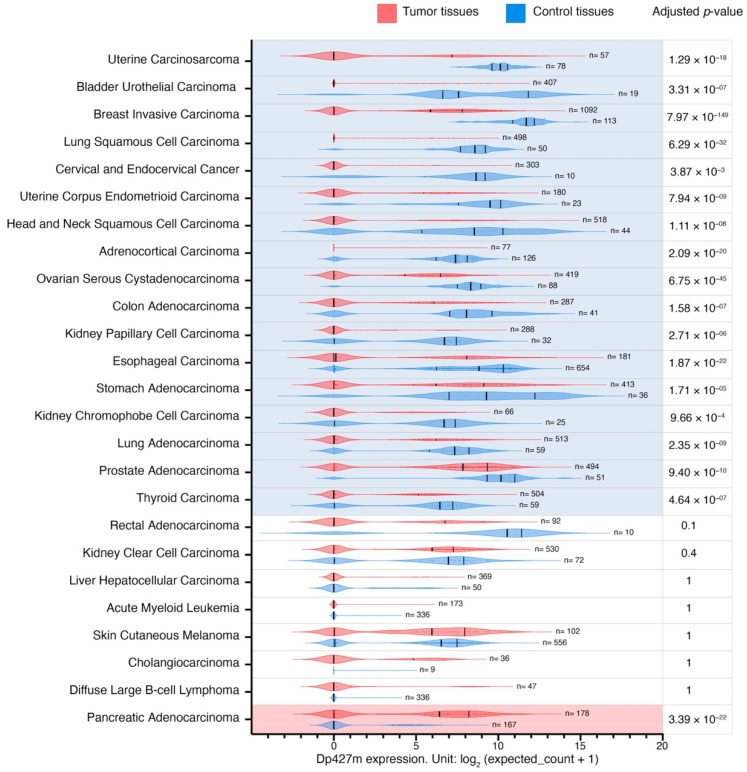
Dp427m transcript expression level in primary tumors and corresponding control tissues. Red and blue violin plots represent tumor and control tissues, respectively. Vertical black lines represent the median and quartiles. Comparisons (n = 17) with a statistically significant downregulation of Dp427m expression in tumor samples compared to controls are highlighted in blue, and one comparison with a statistically significant upregulation of Dp427m expression in tumor samples compared to controls is highlighted in red (*p* < 0.05, Appendix A). The *p*-Values were calculated using the UCSC Xena Browser using a two-tailed Welch’s t test and adjusted for multiple testing using the Bonferroni correction.

**Figure 4 cancers-15-01378-f004:**
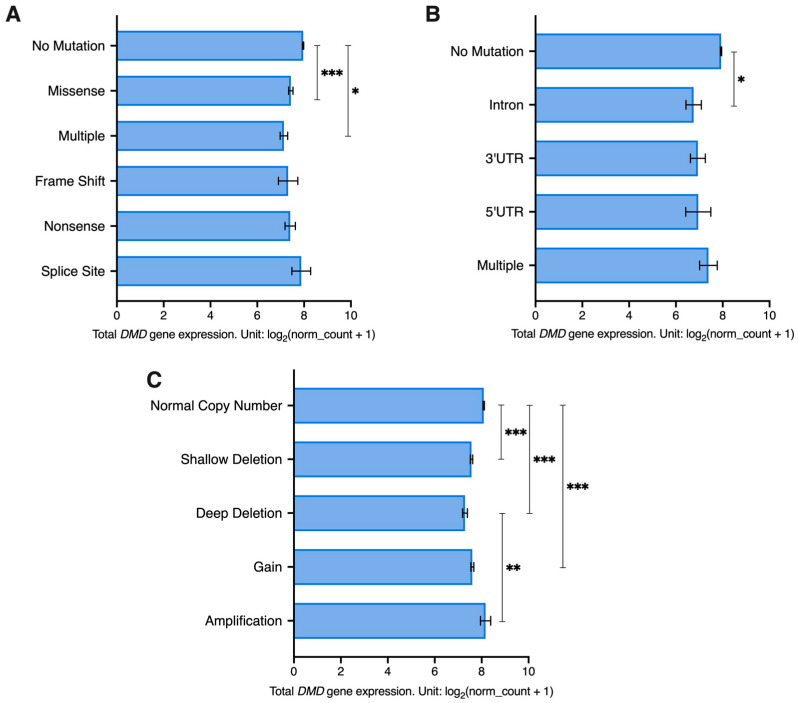
Total *DMD* gene expression in tumor samples from 23 TCGA studies with/without (**A**) CR, (**B**) NCR mutations, and (**C**) SCNAs in the *DMD* gene. The number of samples in each group is indicated in brackets. A deep deletion indicates a deep loss (possibly a homozygous deletion). A shallow deletion indicates a shallow loss (possibly a heterozygous deletion). A gain indicates a low-level gain (a few additional copies, often broad). An amplification indicates a high-level amplification (more copies, often local). Expression data is represented as EM mean ± SEM. The *p*-Values were adjusted using the Bonferroni correction. Asterisks indicate statistical significance (* *p* < 0.05, ** *p* < 0.01, *** *p* < 0.001).

**Figure 5 cancers-15-01378-f005:**
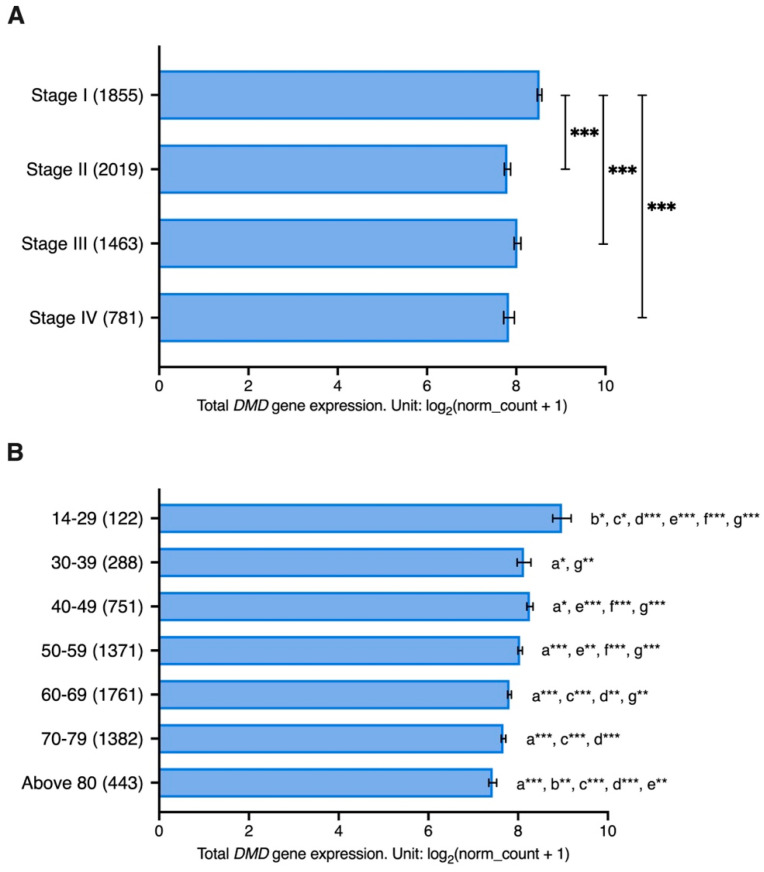
*DMD* gene expression in tumor samples from 18 TCGA tumors with different (**A**) cancer stages and (**B**) age groups. Expression data is represented as EM mean ± SEM. The number of samples in each group is indicated in brackets. The *p*-Values were adjusted using the Bonferroni correction (a: significant difference from 14–29 age group, b: significant difference from 30–39 age group, c: significant difference from 40–49 age group, d: significant difference from 50–59 age group, e: significant difference from 60–69 age group, f: significant difference from 70–79 age group, g: significant difference from above 80 age group, *p* * < 0.05, *p* ** < 0.01, *p* *** < 0.001). Patients aged between 14 and 29 years old had significantly higher *DMD* expression compared to patients in the 30–39 (LogFC = 0.84, *p* = 0.02), 40–49 (LogFC = 0.71, *p* = 0.02), 50–59 (LogFC = 0.93, *p* < 0.001), 60–69 (LogFC = 1.17, *p* < 0.001), 70–79 (LogFC = 1.30, *p* < 0.001), and above 80 age groups (LogFC = 1.54, *p* < 0.001). Patients aged between 30 and 39 years old had higher *DMD* expression compared to patients in the above 80 age group (LogFC = 0.70, *p* = 0.001). Patients in the 40–49 age groups had significantly higher *DMD* levels than patients in the 60–69 (LogFC = 0.45, *p* < 0.001), 70–79 (LogFC = 0.59, *p* < 0.001), and above 80 age groups (LogFC = 0.83, *p* < 0.001). Patients in the 50–59 age groups had significantly higher *DMD* levels than patients in the 60–69 group (LogFC = 0.24, *p* = 0.005), 70–79 (LogFC = 0.37, *p* < 0.001), and above 80 age groups (LogFC = 0.61, *p* < 0.001). Patients aged between 60–69 had significantly higher *DMD* levels than patients in the above 80 group (LogFC = 0.37, *p* = 0.002).

**Figure 6 cancers-15-01378-f006:**
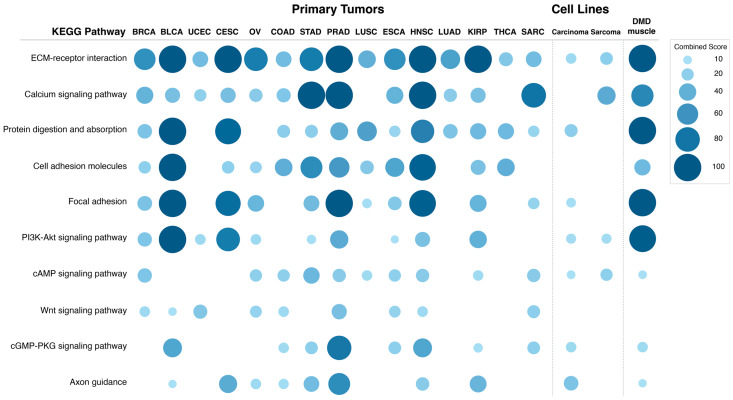
KEGG pathways enriched in the DEGs in primary tumor samples and tumor cell lines with low vs. high *DMD* gene expression as well as in DMD skeletal muscle. Only pathways that were identified in more than 50% of primary tumor samples with low *DMD* expression are displayed (adjusted *p*-Value < 0.05). Data is represented as the combined score for the enrichment (−Log (*p*-Value) × odds ratio). Combined score values greater than 100 were given a value of 100.

**Figure 7 cancers-15-01378-f007:**
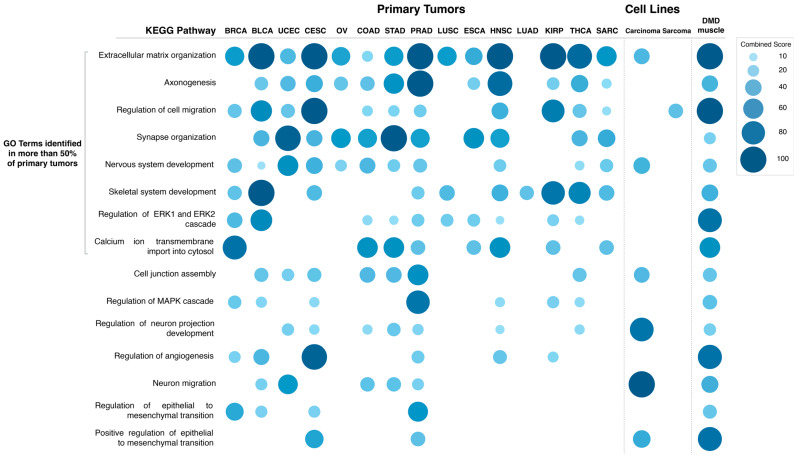
GO Biological Process terms enriched in the DEGs in primary tumor samples and tumor cell lines with low vs. high *DMD* gene expression as well as in DMD skeletal muscle (adjusted *p*-Value < 0.05). Data is represented as the combined score for the enrichment (−Log (*p*-Value) × odds ratio). Combined score values greater than 100 were given a value of 100.

**Figure 8 cancers-15-01378-f008:**
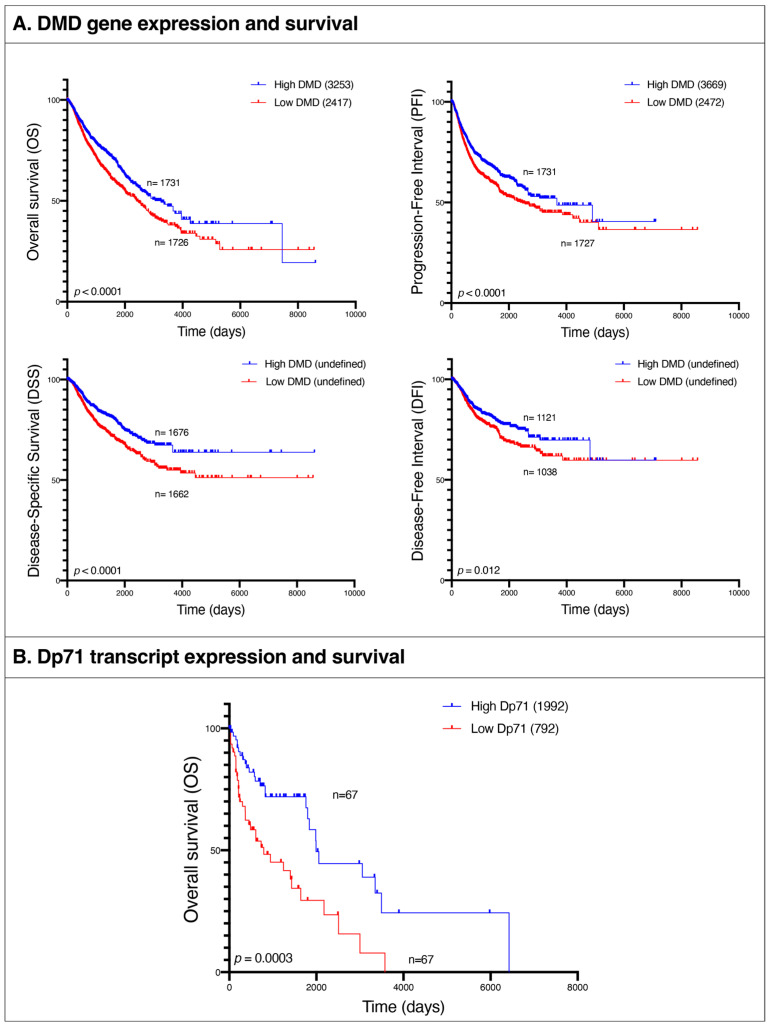
(**A**) Kaplan-Meier curves of overall survival, progression-free interval, disease-specific survival, and disease-free interval for patients with low and high levels of *DMD* gene expression. (**B**) Kaplan-Meier curve of overall survival for hematological malignancies patients with low and high levels of Dp71 expression. Numbers in brackets are overall survival times in days. Numbers of patients in each group and *p*-Values for the log-rank test are displayed in the figure.

**Figure 9 cancers-15-01378-f009:**
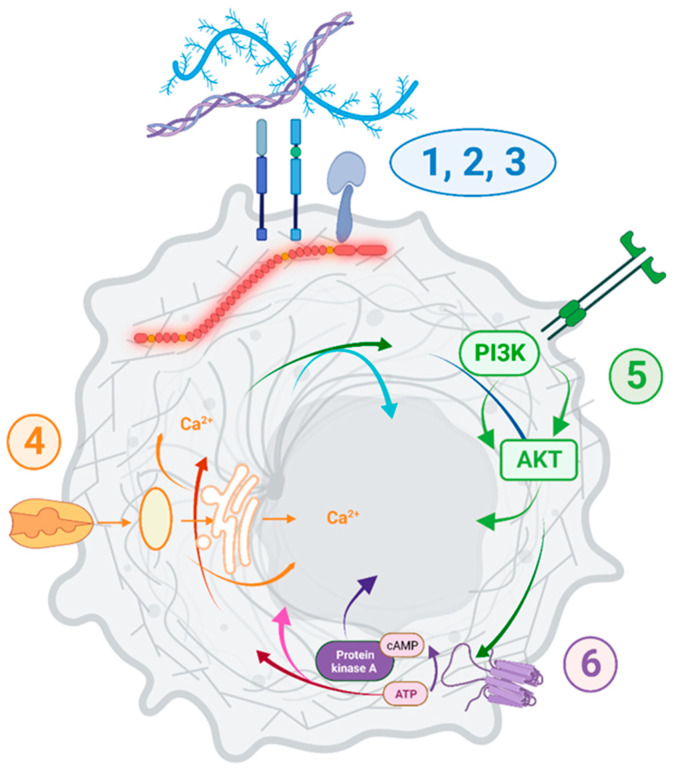
A schematic representation of the key pathways enriched in the DEGs in primary tumors with low *DMD* gene expression that are shared with DMD skeletal muscle. Top six pathways that were identified in more than 50% of primary tumor samples with low *DMD* expression are displayed schematically, including (**1**) ECM-receptor interaction, (**2**) Cell adhesion molecules, (**3**) Focal Adhesion, (**4**) Calcium Signaling, (**5**) PI3K-Akt signaling, (**6**) cAMP signaling. Arrows suggest the potential additional interactions between pathways. Dp427 only is shown.

## Data Availability

All data are presented in the manuscript and Appendix A.

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
