# Peer review of "Downregulation of Dystrophin Expression Occurs across Diverse Tumors, Correlates with the Age of Onset, Staging and Reduced Survival of Patients"

_cancers, 2023, doi:10.3390/cancers15051378_

Round 1

Reviewer 1 Report

This paper has a major missing point: In Duchenne muscular dystrophy where dystrophin protein is substantially reduced, there is no increase in the incidence of any tumors. This reminds me the possibility that may be all these changes are absolutely secondary, which may hold true for some other proteins to be investigated. 

Author Response

This review contradicts the published findings and also appears to miss the main point of our work.

Firstly, recent studies identified an increased frequency of rhabdomyosarcomas in DMD patients (Chandler et al., 2021, Vita et al., 2021), which agrees with the abundance of data from animal models on the increased susceptibility to developing spontaneous rhabdomyosarcomas in dystrophic mice.

Crucially, we demonstrated that downregulation of Dp427 expression occurs across a variety of tumors incidence of which increases with age. DMD patients do not reach the age when these tumors develop. However, with improved therapies, patients will live longer and this risk of malignancy should be considered. As for hematological malignancies, which frequently affect children, poor survival was associated with Dp71 downregulation. This dystrophin isoform is not affected in the majority of DMD patients, and therefore no significant increase in the incidence of such tumors should be expected. We have explained these points in the revised Discussion.

The latter part of this review, dismissing the DMD gene changes in tumors as “absolutely secondary”, is rather vague. We assume that it is to mean that this is a random effect.

While no causation can be confirmed at this stage, DMD dysregulation was associated with specific transcriptomic pathway alterations across 15 primary tumors and 140 various tumor cell lines of different tissue origins. These very same abnormalities are also found in dystrophic muscles, where their causation was firmly linked to the loss of dystrophin expression. Moreover, correlations of lowered dystrophin expression with molecular alterations, earlier age of onset, and decreased survival were confirmed using multiple controls, which excluded random effects.

Reviewer 2 Report

In the present manuscript the authors conduct extensive bioinformatic analysis of past datasets to re-examine the role of the gene products of dystrophin in cancer. Unsurprisingly, given that dystrophin is a component of focal adhesions, and focal adhesions have well known roles in cancer establishment, metastasis, and lethality, they find dystrophin appears to contribute to these past observations. Specifically, figures 6 and 7 highlight that dystrophin's role in ECM, Calcium signaling, Insulin/PI3K/Akt signaling (and of course focal adhesion) appear to be key drivers (and of course are some of the key drivers of FA and CA). Overall the manuscript is very well put together, particularly the focus on dystrophin levels and dystrophin isoforms. The main detractor, for me, was not fully embracing the past literature and instead focusing on the "bias" against dystrophin being important for anything other than DMD (which is a strong clinical and noticeable bias in pre-clinical).

Minor points:

1) Simple summary- dystrophin (as pointed out in your discussion) is in fact known to be expressed in most tissues.

2) I might focus less on "illegitimate transcription" and more on the novelty of your findings (esp. as human protein atlas has previously demonstrated that dystrophin is universally detected in CA cell lines and most tumors).

3) Abstract- "developmental onset of DMD" probably should be removed- it is not fully explained and is largely not relevant to the present study.

4) Introduction- again it is not really true that dystrophin is not expressed in these tissues. Might remove this statement or say not widely believed to be expressed (which is different from the objective truth)

5) Results 3.3. - The non computationalist (e.g. biologist/clinician) will want to known what the link between the three clusters, Dp427m, and gene expression is. You might provide some forward link to this by referencing that this link will be explored in section 3.7.

6) section 3.5 is probably quite important for the cancer field where "oncogenes" are a key concept. You might highlight this more in the discussion.

7) Discussion- seems a bit long and unfocused. You might consider highlighting why Human protein atlas did not find cancer specificity or prognostic value (e.g. it is fairly universal and it abundance not presence/absence that matters) as well as it really is gene expression and not somatic mutation and also highlight the gene expression changes/regulation and make the parallel to FA roles in CA.

That said, as a whole, I feel this work is good so do not feel obligated to adopt any of my minor comments.

Author Response

We very much appreciate the very positive comments and constructive suggestions. While reporting the findings on the role of the DMD gene products in cancer, we stressed the similarity in pathways altered in a diverse range of tumours and Duchenne.  We would not consider these commonalities unsurprising. Although alterations in ECM interaction and calcium signalling pathways occur in dystrophic muscle and these alterations have well-known impacts in cancer, it was rather unexpected to find these pathways altered in tumours derived from tissues (epithelia, blood) in which DMD expression was not detected or disregarded. Therefore, we indeed stress these aspects of our findings in order to counter this long-term bias of focusing only on dystrophin in myofibers when investigating the pathology and treatment of DMD.  

Minor points:

RE: 1. While dystrophin was known to be expressed in most tissues, its significance was disregarded, and we are not aware of any other studies identifying DMD as a housekeeping gene.

RE: 2. As we described, Human Protein Atlas gives incomplete if not misleading information.  Therefore, as explained earlier, we consider it important to emphasise that the widespread dystrophin expression is functionally relevant.

RE: 3. We believe that the developmental onset of DMD is directly relevant to this study. It has been described in DMD foetuses decades ago, but its molecular underpinnings investigated recently (Mournetas et al., 2021). The idea for this study stems from the fact that multiple mechanisms in embryogenesis and carcinogenesis are shared. We hypothesised that dystrophin alteration will evoke related outcomes.

RE: 4. This statement has been modified, as suggested (line 86).

RE: 5. We very much appreciate this suggestion and modified the manuscript accordingly Lanes 423 and 424).

RE: 6 and 7. We attempted to highlight these points by restructuring the Discussion.

Reviewer 3 Report

The authors investigated in this research paper Dystrophin expression levels in different healthy tissues and their cancer counterparts and then correlated expression levels of DMD in cancer with gender, age, and overall survival of patients.

The paper is nicely written, very clear, and of interest even though it is a correlative work, the results highlight and sustain the conclusion that DMD would have a role in tumorigenesis.

I believe that it would be beneficial for the readers to have a global table summarizing the various correlations made in the study including the various parameters that were studied.

I also think that an additional figure showing at the cellular levels that DMD is at the interconnection of a network regulating various cellular mechanisms (migration, calcium homeostasis..) will be highly appreciated by the readers.

As a minor comment, authors should homogenize tumors or tumours for instance both are found in the simple summary

Author Response

We very much appreciate the very positive comments and constructive suggestions.

We have added a figure (Fig. 9) in which we attempted to illustrate these interconnections in a schematic fashion. However, we found it difficult to create a summary table that would be illustrative and yet not detailed enough to resemble tables already included in Supplementary File 1.

This spelling inconsistency has now been rectified.